# The Molecular Mechanisms of Adaptive Response Related to Environmental Stress

**DOI:** 10.3390/ijms21197053

**Published:** 2020-09-25

**Authors:** Andrea Rossnerova, Alberto Izzotti, Alessandra Pulliero, Aalt Bast, S. I. S. Rattan, Pavel Rossner

**Affiliations:** 1Department of Genetic Toxicology and Epigenetics, Institute of Experimental Medicine, 14220 Prague, Czech Republic; andrea.rossnerova@iem.cas.cz; 2Department of Experimental Medicine, University of Genoa, 16132 Genoa, Italy; izzotti@unige.it; 3IRCCS Ospedale Policlinico San Martino, 16132 Genoa, Italy; 4Department of Health Science, University of Genoa, 16132 Genoa, Italy; 5Department of Pharmacology and Toxicology, Maastricht University, 6200 MD Maastricht, The Netherlands; a.bast@maastrichtuniversity.nl; 6Campus Venlo, Maastricht University, 5900 AA Venlo, The Netherlands; 7Department of Molecular Biology and Genetics, Aarhus University, 8000 Aarhus, Denmark; rattan@mbg.au.dk; 8Department of Nanotoxicology and Molecular Epidemiology, Institute of Experimental Medicine, 14220 Prague, Czech Republic; prossner@biomed.cas.cz

**Keywords:** adaptive response, preventive medicine, microRNA machinery

## Abstract

The exposure of living organisms to environmental stress triggers defensive responses resulting in the activation of protective processes. Whenever the exposure occurs at low doses, defensive effects overwhelm the adverse effects of the exposure; this adaptive situation is referred to as “hormesis”. Environmental, physical, and nutritional hormetins lead to the stimulation and strengthening of the maintenance and repair systems in cells and tissues. Exercise, heat, and irradiation are examples of physical hormetins, which activate heat shock-, DNA repair-, and anti-oxidative-stress responses. The health promoting effect of many bio-actives in fruits and vegetables can be seen as the effect of mildly toxic compounds triggering this adaptive stimulus. Numerous studies indicate that living organisms possess the ability to adapt to adverse environmental conditions, as exemplified by the fact that DNA damage and gene expression profiling in populations living in the environment with high levels of air pollution do not correspond to the concentrations of pollutants. The molecular mechanisms of the hormetic response include modulation of (a) transcription factor Nrf2 activating the synthesis of glutathione and the subsequent protection of the cell; (b) DNA methylation; and (c) microRNA. These findings provide evidence that hormesis is a toxicological event, occurring at low exposure doses to environmental stressors, having the benefit for the maintenance of a healthy status.

## 1. Introduction

Environmental stresses are present for all living beings. The term stress can be used to refer to the organism’s response to a stressful stimulus, or to the consequences of this response [1]. Organisms adapt to stress through defined regulatory mechanisms that drive changes in gene expression, body morphology, and physiology thus triggering the defensive response. Consequently, it is very important to understand how cells and tissues react to stress in order to survive such threats. Stress-response defensive mechanisms can be modulated and activated in healthy organisms by preventive interventions including lifestyle, food, and administration of chemopreventive natural principles or drugs. As an example, several epidemiological and human intervention studies have been carried out on the protective effects of foods (polys)phenol-rich in phenol against different chronic diseases, including neurodegeneration, cancer, and cardiovascular diseases [2]. However, also environmental exposures, either physical or chemical, trigger a variety of defensive mechanisms in the exposed organisms. This situation is referred to as “adaptive response”. Whenever the amount, intensity, and duration of the exposure overwhelms the defensive machinery, the exposure results in a risk for health. The hallmarks of aging include the accumulation of genomic damages, epigenetic alterations, the loss of proteostasis, and deregulated nutrient sensing [3]. Indeed, the aging process is affected by both genetic factors and epigenetic mechanisms, that are potently correlated with each other [4]. For example, environmental cues such as nutrient intake can interact with DNA structures and alter transcriptional profiles, which could elicit stable changes in the aging of the organism. The exposure to low doses of environmental agents result in an environmentally induced modification in the phenotype that displays an enhanced adaptive response to the consequent higher dose [5].

Epigenetic changes (DNA methylation, microRNAs, and histone acetylation) and transcriptional silence with miRNA, permanently affect the reading of genes. Environmental factors interact with genome and gene transcription modulating the epigenetic machinery [6]. This adaptive epigenetic arrangement, starts since the early stages of the development of the organism during pregnancy [7], blows up at the delivery of newborns in terms of oxidative stress targeting the lung [8], proceeds during infancy and adulthood [9], and overwhelms the defensive machinery of the organism during aging [10]. These arrangements are the cumulative result of exposure to low doses and the resulting hormetic reactions [11]. Whenever the amount, intensity, and duration of the exposure is well below the capacity of the defensive machinery, the exposure only activates defensive mechanisms that remains active to defend the organisms against further exposure without any risk for health damage. This situation is referred to as “hormesis”. This situation has been reported for environmental exposures to numerous oxidants, exciting radiation, hypoxia, and stressful procedures [12]. The environmental exposures could produce DNA mutations representing a major landmark for risk assessment and prevention [13]. Epigenetic alterations are relatively stable throughout life and are linked to different biological processes, health, and diseases [14]. Epigenetic regulation of gene expression is a fundamental molecular mechanism that links environmental factors with the genome. Indeed, epigenetic changes are much more frequent than genetic changes, and many of these changes are adaptative. Recent experimental studies indicate that environmental fluctuations can lead to changes in which adaptive responses to low doses of hazardous conditions improve the functional ability of cells and organisms. As an example, the exposure to environmental ionizing radiation selects defensive genotype polymorphisms that results more frequently after three generations thus attenuating the consequences of this environmental exposure [15]. Environmental, physical, and nutritional hormesis lead to the stimulation and strengthening of the maintenance and repair systems in cells and tissues. Physical activity blows up endogenous oxidative stress that, as rebound adaptive response, triggers the long-term activation of antioxidant defenses as demonstrated for antioxidant availability in blood [16]. Since hormesis appears to be a relatively common phenomenon in many areas, the objective of this review is to explore its occurrence related to low exposure doses to environmental stressors, having the benefit for the maintenance of a healthy status.

## 2. Hormesis for Healthy Ageing and Longevity

One of the research areas where the concept of hormesis is widely accepted and applied is in modulating ageing and longevity of cells and organisms [17], and is based on the fact that the adaptive behavior of biological systems in response to environmental or self-imposed mild stress(es) improves their functionality and survival. Physical, nutritional, and mental stresses or challenges which induce hormesis, termed hormesis, lead to the stimulation and strengthening of the maintenance and repair systems in the body [17]. Some examples of physical hormesis are exercise, heat, and irradiation, which activate anti-oxidative, heat shock, and DNA repair-stress responses, respectively [18]. A wide variety of non-chemical components in the food, such as flavonoids and polyphenols present in spices, herbs, and other sources, are examples of nutritional hormesis, which induce anti-oxidative, anti-inflammatory, and autophagy stress responses. Similarly, calorie restriction (CR) and intermittent fasting are also hormesis, which activate the autophagic and sirtuin-mediated stress responses [19].

CR appears to prolong life by modulating reactive oxygen species (ROS)-mediated oxidative damage through ROS formation, which is a highly regulated process controlled by a complex network of intracellular signaling pathways [19]. Furthermore, the nuclear factor erythroid 2-related factor (Nrf2) binding to antioxidant response elements (AREs), regulates the basal and inducible expression of glyoxylase 1 (Glo1), as well as of AKRs and ADH [20]. Reduced activity of Nrf2 and increased oxidative stress in aging and disease may predispose to dicarbonyl stress, which is beginning to feature strongly as a driver of pathogenesis in aging-related disease. In a similar vein, intracellular nutrient and energy status, the functional state of mitochondria, and the concentration of ROS produced in mitochondria are involved in the regulation of lifespan across species by coordinating information and divergence of multiple branched signaling pathways, including vitagenes in preserving cellular homeostasis during stressful conditions [21]. Intense brain activity and focused attention comprise mental hormesis, which also induce various stress responses, including heat shock response. In a similar vein, intracellular nutrient and energy status, the functional state of mitochondria, and the concentration of ROS produced in the mitochondria are involved in the regulation of lifespan across species by coordinating information and divergence of multiple branched signaling pathways, including vitagenes in preserving cellular homeostasis during stressful conditions [21]. Intense brain activity and focused attention comprise mental hormesis, which also induce various stress responses, including heat shock response [21].

An important characteristic of hormesis for health is the simultaneous stimulation of many independent cellular functions/endpoints—each with its own set of quantitatively hormetic features. For example, enhancements of DNA repair, antioxidant defenses, autophagy, etc., whose actions are regulated by multiple interacting receptor/signaling pathways, ultimately produce a metabolically integrated and coherent cellular response [20]. More importantly, the hormetic response has specific characteristics which define both the quantitative features of biological plasticity and the potential for maximum biological performance, thereby estimating the limits to which numerous medical and pharmacological interventions may or may not affect humans [20]. Therefore, a combination of different hormesis can be the drugs for maintaining, improving, and recovering health during aging [17,18].

## 3. Biomarkers of Adaptive Responses in Human Health

WHO defined health as a state of complete physical, mental, and social well-being [22]. Today, there is a more dynamic definition of health, that is “the ability of an organism to adapt to the environment” [23].

Adaptive responses largely explain the health benefits of fruits and vegetables [9]. Indeed, many natural chemopreventive agents, are detoxified by the phase I/phase II metabolic reaction thus activating the involved enzymes and regulating pathways [24]. As an example, this situation typically occurs for indole-3-carbinole [25] and catechins [26]. However, nowadays it is increasingly recognized that also environmental toxicants frequently display an hormetic response. This has immense consequences in risk assessment [27]. We now understand some molecular mechanisms of this hormetic response. Incubation of lung epithelial cell with a low concentration of acrolein leads to activation of the transcription factor Nrf2 [28]. This activates the synthesis of glutathione and the subsequent protection of the lung cells to a high concentration of acrolein [29]. Moreover, a low dose of silver nanoparticles has been shown to activate Nrf2 and thus to hormesis [30].

This background knowledge on the mechanistic aspects of hormesis enables us to define specific biomarkers to follow this process [31]. 

Accordingly, hermetic biomarkers depend on the specific mechanisms triggered by the hormetic condition considered and may be either genetic, epigenetic, or metabolic. Genetic biomarkers include the decrease of genotoxic damage as evaluated by DNA adducts or cytogenetic biomarkers [32]. Epigenetic biomarkers mainly include miRNA due to their specific and important role in triggering and regulating the early stages of the adaptive response [33]. 

## 4. Epigenetic Aspects of Human Adaptation to Environmental Pollution

Although results of numerous studies indicate that living organisms possess the ability to adapt to adverse environmental conditions, mechanisms of adaptation are not well understood. It is generally believed that the induction of repair systems and antioxidant response activated as a result of contact of the organism with environmental pollutants during life play a major role. However, there is a strong indication that the prerequisite for adaptive response that occurs later in life may already develop during the prenatal development and that epigenetic mechanisms, namely DNA methylation as a regulatory element of gene expression, play an important role. 

The results of our studies proved that DNA damage and whole genome gene expression profiling in populations living in the environment with high levels of air pollution do not correspond to the concentrations of pollutants. Moreover, DNA damage levels associated with air pollution were affected by the place of birth of the study subjects and were higher in those who were born in clean localities with low air pollution levels. Analyses of methylation profiles of children living in localities differing in environmental pollution levels showed clear clustering depending on the place of residence. DNA methylation was further affected by various past events including those that occurred during gestation and shortly after birth (length of gestation, birth weight, length of breastfeeding). These findings indicate the existence of epigenetic memory that is set during the prenatal development and affect the response of the organism to environmental conditions later in life. Several studies have shown that environmental exposures are very relevant during embryonic development where in these periods there is an epigenetic reprogramming of the offspring. Epigenetic alterations in the germ line are relevant as they can be transmitted trans-generationally and could be associated with different reproductive disorders, as demonstrated for the endocrine disruptor vinclozolin [34].

### 4.1. The Role of DNA Methylation in Epigenetic Adaptation

#### 4.1.1. Basic Strategies of Adaptation

Adaptation, defined as the ability “to adapt to a new situation with the aim to increase the chance or quality of surviving”, uses two basic strategies [35]. 

First of them “genetic way of adaptation” is associated with slow processes, such as induction of mutations or selection of a specific set of genetic polymorphisms. This strategy is characteristic for long-term stressor exposure in isolated populations. Even though this process is permanent, it’s very slow speed that may take many generations, is a major disadvantage especially for modern-day human populations whose lifestyle is characterized by frequent environmental changes. An interesting example of this mutation strategy is the induction of mutation (a six-base deletion) in aryl hydrocarbon receptor 2 gene (*AHR2*) in tomcods exposed to polychlorinated biphenyl (PCB) in the Hudson River [36,37]. Another example associated with the selection of a specific genetic polymorphism pattern in a native human population from Argentina exposed to the high concentration (150/200 µg per L) of arsenic in drinking water was reported in 2015. This population, studied already in 1995 and 1996, showed no increase of cytogenetic markers related to such high exposure of arsenic. The unique quick metabolism of detoxification was suggested as a reason for this observation. A detailed analysis published about twenty years later identified 13 specific single nucleotide polymorphisms (SNPs) in the arsenic methyltransferase gene (*AS3MT*) [38,39,40,41].

The second strategy, a more dynamic and relatively short-term process “epigenetic way of adaptation” is represented by rearrangement of the epigenetic pattern especially in DNA, as well as directly in histones. This solution leads to changes in the intensity of gene expression at two levels: (i) Intensity of transcription that is affected especially by the level of DNA methylation in promotor regions of both protein-coding and noncoding genes, and (ii) intensity of translation that can be consequently regulated via miRNAs molecules. The dynamicity of this process allows adaptation to new chronic environmental stressors within a short period of time, even during the individuals’ life. This is the main benefit of this strategy (some examples from human biomonitoring studies are presented in Section 4.1.2).

#### 4.1.2. Epigenetic Way of Adaptation and Various Environmental Stressors

Each human is exposed to numerous environmental stressors during life, even in the prenatal period of the development. Probably the most common exposure which we cannot easily affect is associated with specific air pollution related to our residence. In contrast, other exposures, especially those specific for occupation and lifestyle, could be partly modified by each of us. Up to now, numerous studies reporting epigenetic changes in the DNA methylation pattern related to chronic exposure have been published.

*Air pollution:* Probably the most complex research that summarized data from a decade long investigation in three cohorts exposed to various types and levels of air pollution has been reviewed recently [42]. The data showed that in some studies no effects or even positive response after chronic air pollution exposure were observed [43,44]. In contrast, subjects with acute exposure, which temporarily moved from the environment with low levels of air pollution to highly polluted regions, exhibited a significant increase of DNA damage [45]. These observations may be explained by analyses of molecular biomarkers: (i) Cohorts from various locations differed in gene expression levels, including changes in DNA repair genes [46,47], and (ii) substantial differences in the DNA methylation pattern were observed between the groups from a rural and industrial area [48]. Moreover, the role of exposure levels in the prenatal period related to sensitivity to the environmental stressor in later life, including effects on DNA damage levels was suggested [42]. This was probably the first attempt to explain and suggest a versatile epigenetic basis of adaptation related to the environmental exposure in such complex studies evaluated using both traditional and molecular biomarkers.

In addition to this research, many other studies reporting differences in DNA methylation after exposure to traffic related air pollution, particulate matter, black carbon, ozone, nitrogen oxides or polycyclic aromatic hydrocarbons were published (summarized in recent review articles) [49,50]. These works documented epigenetic changes and a specific DNA methylation pattern in individual cohorts exposed to the same environmental stressors in one location (the same type of air pollution exposure). This fact is crucial, as pooling such different cohorts may result in a limited number of overlapping epigenetic patterns in individual cohorts. This is in agreement with a meta-analysis of epigenome related to NO_2_ and NO_x_ exposures in various cohorts that did not show genome-wide significant associations at single CpG site level [51].

*Occupational exposure:* A similar, currently ongoing, complex study, is related to the occupational exposure to nanoparticles. Here, nanocomposite research workers and matched controls were repeatedly (annually in years 2015–2019, twice per day: Before and after exposure) monitored. Both, cytogenetics and epigenetics approaches were involved together with nanoparticles exposure monitoring and composition measurements. The design of the study allowed analyzing both the effect of chronic exposure (years), and acute exposure (short-term daily exposure associated with specific activities in the workshops on a given day). Similar to the air pollution studies mentioned above, no effect of chronic exposure on the frequency of total micronuclei was observed [52,53], unlike a significant increase related to acute exposure. These data were in agreement with epigenetic data that showed no differences in the methylation pattern after acute exposure, but induction of different methylation of 705 CpG loci in the exposed subjects after 14.5 years of chronic exposure [54]. These results indicate again the epigenetic adaptation to the environmental stressor with consequences for the decrease of DNA damage. Interestingly, the changes were less pronounced than those observed in the air pollution methylation study supporting the fact that unlike air pollution the occupational exposure is not a permanent environmental stressor.

Other data showing changes in DNA methylation patterns related to occupational exposure were reported for firefighters [55], or subjects exposed to pesticides [56,57], or metals [58]. A general summary with more evidence of effects of environmental chemicals on epigenetic regulation in various studies (human, animal, and in vitro) including perspectives for the future of the field of toxico-epigenomics were recently published by the epigenetic group of International Agency for Research on Cancer (IARC) [59].

*Lifestyle factors:* Lifestyle factors are additional elements which can affect the epigenetics settings including the DNA methylation pattern. Most of these factors can be influenced by each of us during life, but some of them could result from the global unfavorable conditions or exposure during the prenatal development. Proofs of DNA methylation changes in offspring were observed in relation to the mother’s famine during pregnancy in the Dutch Hunger Winter in 1944–1945 [60]; mother’s alcohol consumption [61]; or mother’s smoking [62].

The effect of smoking on the methylation pattern during life is probably the most frequently studied and confirmed lifestyle factor. Moreover, one study reported faster dynamics of DNA methylation changes in former smokers who started smoking again than in non-smokers [63], which implicates the manifestation of epigenetic memory (more details in Section 4.1.3). The recent studies focus also on the modulation of the epigenetic landscape (including DNA methylation, histone modification, and non-coding RNA such miRNA) in the context of exercise-related adaptation [64].

#### 4.1.3. Epigenetics Memory as a Tool for Keeping of Previous Adaptation Settings

The previous section (Section 4.1.2) showed numerous examples of the effect of environmental exposure on the epigenetic modification via the DNA methylation pattern. Among them, some results indicated that the epigenetic changes were memorized by the cells and could play an important role in adaptation in the case of future re-exposure. Moreover, the predictive adaptive response (PAR) via epigenetic mechanisms was described in relation to prenatal exposure as a tool, which can help modify functions of the genes to this exposure as an anticipated environment in later life [65]. Not only exposure in prenatal development, but also another exposure episode during the later life, especially low dose chronic exposure, can be a reason for epigenetic adaptation of our genome.

A question remains how the epigenetic adaptation is kept during the many years of life. Most of the proofs of epigenetic differences were shown in the blood cells, but their life span is limited and thus they cannot be bearers of epigenetic memory. Despite that, the existence of epigenetic memory was previously described as a mechanism which allows the organisms to adapt to environmental changes [66,67]. Recent hypotheses suggest that stem cells play a significant role in keeping adaptation settings. Information about the epigenetic memory in relation to reprograming and differentiation processes in induced pluripotent stem cells (iPSC) were published already in 2010 [68]. Even though the “plans” for differentiation are the main priority of stem cells, there is evidence, that the environmental exposure can also affect their epigenome. Examples of these alterations include, e.g., changes in 5-hydroxymethylcytosine clusters in CpG islands in human embryotic stem cells [69] or accumulation of DNA methylation alterations in pediatric glioma stem cells following fractionated dose irradiation [70]. The hypothesis of epigenetic (methylation) memory was recently proposed as an analogy to immune memory. According to this concept, the CpG methylation pattern in specific genes persists in stem cell compartments as a form of adaptive epigenetic memory. The respective stem cells can then be activated and multiplied by clonal expansion when the organism reencounters the toxicant or other stimulus [71]. This process perfectly fits the explanation of some previously published retrospective data, e.g., differences in the DNA methylation pattern related to the length of full breastfeeding observed in children aged 7–15 years; or rapid manifestation of adaptation processes related to DNA methylation changes in former smokers; or persistent changes of CpG islands in gene promoters of blood cells detectable in humans exposed to ionizing radiation long time ago (2–46 years) [48,63,72].

Based on the above-mentioned published data, we propose a model of the process of adaptation and its “storage” by epigenetic memory in stem cells (Figure 1; description is provided in the following text). The model is highly simplified, as each of us is chronically or acutely exposed to low doses of numerous chemicals during life. In the model, we show several virtual situations of chronic/acute exposure/re-exposure during adulthood and describe the consequences related to the risk of DNA damage and forming of epigenetic changes via the DNA methylation pattern including their “storage” via stem cells. The “story of our virtual person and virtual environmental exposure” begins in an adult person aged around 20 years (Figure 1a). A new chronic exposure A affects the epigenome modification of stem cells, but the originally unmodified epigenome of blood cells does not protect them from DNA damage (typical reaction after acute, short-term exposure) a short time after exposure starts. However, this is not detectable later during chronic exposure and in post-exposure time A, because the DNA of stem cells is already modified, and new blood cells contain this altered epigenetic pattern. Later, the same person (now around 30 years old) is chronically exposed to a different chemical (chronic exposure B, Figure 1b). The scenario of processes is now identical to exposure A with the only difference that the pre-exposure epigenome of both stem cells and blood cells contains epigenetic modification for exposure A, which was memorized, but is not effective for exposure B. A new epigenetic modification is set and stem cells and blood cells are now epigenetically modified and adapted to exposure A and B. Again later, the same person (now around 40 years old) is acutely exposed to chemical C (acute exposure C, Figure 1c). The epigenome of both stem cells and blood cells remains without any new modification and contains changes induced by exposure A and B, but not C, as it was too short. In blood cells, DNA damage is temporarily increased, similar to early stages of chronic exposure. In the post-exposure period, DNA damage is again reduced, but the cells are not adapted to exposure C and no modification is kept in epigenetic memory. Another exposure episode that affects our virtual person, now around the age of 50 years, is the same chronic exposure as that occurred 30 years ago (chronic re-exposure A, Figure 1d). As shown in the figure, stem cells, as well as blood cells still contain epigenetic changes induced by the original exposure A and as a result, DNA damage is not increased. The already adapted cells manifest their epigenetic pattern stored for decades in epigenetic memory and cells effectively prevent possible DNA damage. A similar situation occurs in our person now aged around 60 years after acute re-exposure B (acute re-exposure B, Figure 1e). Even though this time the exposure was only short-term, the previous epigenetic modification after exposure B allowed avoiding a possible DNA damage. We can conclude our story by a statement, that our epigenome is like a book with chapters about our previous exposure stories which can be re-read in the case of repeated exposure in the future.

In the model shown in Figure 1, we explicitly focus on the situation during adulthood, but it is important to mention that processes of epigenome adaptation begin already in prenatal development. We can presume that the process is even more effective and faster during this period and early childhood, due to the massive division of cells during prenatal development and growth, which can spread the new epigenetic setting faster into more cells in the body. Based on these facts, we can presume that the process of epigenetic adaptation is significantly slower in later life as aging is linked with the slow process of regeneration.

#### 4.1.4. Health Effects of Exposure vs. Processes of Epigenetic Adaptation

The processes of epigenetic adaptation can be difficult to understand and even unacceptable for many researchers as there are a huge number of studies (more than 80,000 in the PubMed database) reporting health effects outcomes directly linked to environmental exposure [73,74,75,76]. Some studies presented direct health effect consequences after acute exposure or early stages of chronic exposure [73]. In contrast, chronic exposure in areas with long-time “environmental pressure” could produce significant health effects in the future [75]. Especially this observation related to chronic exposure represents a direct link between the process of epigenetic adaptation and the health effect risks in later life. The ability of the epigenetic adaptation, in connection with new settings of the functions of genes in new conditions, is very important for survival of populations in unfavorable conditions and represents the major advantage of this process. On the other hand, changes in expression of some genes could cause significant health effects in the future. Both factors, epigenetic adaptation/modification and health effects affected by our exposure history, should be studied concurrently as particular epigenetic changes are connected with concrete health effect risks in the future. Finally, the epigenetic adaptation is a mechanism for reduction of more serious health effects induced in populations during chronic exposure. It helps delay or reduce the onset of negative reproductive outcomes [76,77]. An overview of triggered mechanisms of hormesis is reported in Table 1.

The role of epigenetic changes in adaptation to the environment was hypothesized and reviewed in the literature during the last few years. Table 2 summarizes the key reviews or hypotheses and their conclusions concerning these aspects.

## 5. The Role of microRNA Machinery in the Adaptive Response

MicroRNAs are regulators of gene expression at the postgenomic level. This regulation has a profound impact on the body’s early response to exposure to environmental carcinogens. At the beginning of exposure, the microRNA mechanism reacts promptly by changing it mainly in the sense of selective downregulation, thus allowing the expression of gene coding for phase I/II detoxification activities, as well as DNA/protein repair [83]. Under physiological conditions, there is no overlap between the intensity of expression of mRNA and related proteins for genes encoding for phase I/II detoxification and DNA/protein repair activities. Indeed, microRNA blocks at the post transcriptional level the translation into proteins of defensive activities not needed in an unexposed organism, thus allowing energy and metabolite saving. Conversely, when the organism is exposed to genotoxic carcinogens, such as hexavalent chromium, blocking microRNA are silenced thus allowing the translation of mRNA into proteins activating phase I/II detoxification and DNA/protein repair activities [84]. Under this perspective, microRNAs are the first line of interception against xenobiotics. The microRNA machinery displays various mechanisms to intercept xenobiotics. Indeed, microRNA has been initially developed billions of years ago in plants to intercept and destroy foreign nucleic acids. Mammals have extended this approach also to xenobiotics. Stress-sensitive pre-microRNA in cell cytoplasm are enriched by nucleophilic G in their terminal loop. This nucleophilic site can intercept electrophilic compounds as the activated metabolites of genotoxic agents characterized by high electrophilicity, an attribute required for DNA binding. The results of this interaction are the formation of a xenobiotic-microRNA adduct that cannot be further processed to mature microRNA by DICER, being unable to penetrate the catalytic pocket of this enzyme. This situation occurs in the case of short-term exposure and results in the activation of an adaptive response protecting DNA organisms from xenobiotics. No damage occurs; indeed xenobiotic-microRNA can be easily removed or extruded outside the cells and promptly restored by newly synthesized pre-microRNA.

Moreover, the DICER surface, in the area surrounding its catalytic pocket, is composed of weakly nucleophilic amino acids. This stereochemical situation allows the correct orienteering of the substrate, i.e., the pre-microRNA that is pushed in the deep of the catalytic pocket for its final processing. In the case of long-term exposures to xenobiotics, their nucleophilic metabolites progressively accumulate onto the nucleophilic amino acids composing the wall of DICER catalytic pocket. This situation results in a progressive loss of function of DICER. Unfortunately, DICER is an extremely complex tetrameric enzyme that, at variance with pre-microRNA, cannot be replaced or repaired. Accordingly, the long-term exposure results in an irreversible blockage of the microRNA maturation machinery [85]. DICER blockage by xenobiotics is an unspecific event targeting many microRNAs also including those suppressing the expression of mutated oncogenes. Accordingly, the long-term exposure irreversibly suppresses the microRNA function thus disclosing the phenotypic expression of mutated oncogenes triggering the carcinogenesis process [86].

These molecular adaptive mechanisms explain why the population exposed to low doses of environmental agents adapts well to the adverse environmental situation with only minimal alteration of their molecular damage biomarkers [42]. However, whenever exposures persist for many decades, they are a major risk factor for the appearance of cancer. 

## 6. Conclusions

The existence of the hormetic effect in environmental toxicology has remarkable consequences in preventive medicine and environmental hygiene. Since hormesis occurs only at low exposure doses, there is no doubt that all the ongoing efforts to reduce pollutants in the environment are absolutely worthy to be pursued. However, the final goal is not the environmental zero dose, that is often an utopic goal for many pollutants generated from natural sources or existence means that, at least for the environmental toxicants for which this event is well established, low doses can be tolerated. The quantification of this “low dose” is extremely difficult because of the inter-individual variability in sensitivity to health effects of environmental pollutants. Indeed, fragile subjects (e.g., aged subjects, children, fetuses) having poor inducibility of their defensive mechanisms activated by hormesis, can receive health risk by lower exposure doses than doses tolerated by other subjects. 

Accordingly, hormesis has relevance in preventive medicine as a tool that is able to enhance endogenous defenses by correct nutrition (chemopreventive functional foods) and healthy lifestyle (e.g., physical activity). This approach, paralleled by the progressive decrease of the amount of pollutants in the environment will allow the avoidance of health risk well before the reaching of a zero dose of pollutants in the environment.

## Figures and Tables

**Figure 1 ijms-21-07053-f001:**
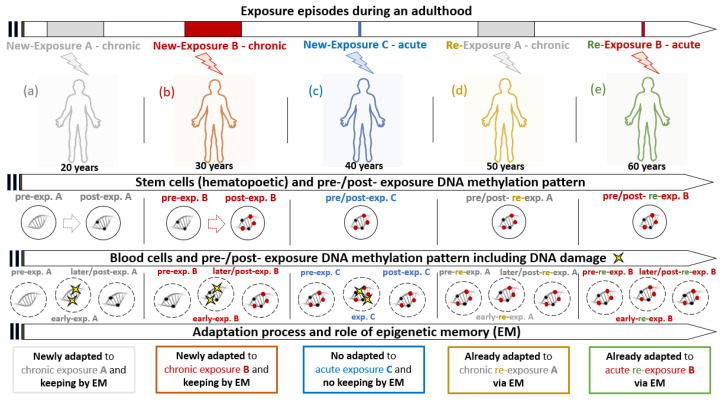
The model of the process of adaptation to environmental exposure and its “storage” by epigenetic memory in stem cells (see Section 4.1.3 for a detailed description of consequences of individual exposure episodes **a**–**e**).

**Table 1 ijms-21-07053-t001:** Hormesis: Practical examples and triggered mechanisms.

Exposures	Mechanisms	Biological Effect
*Environmental*		
Low airborne pollution	- microRNA - phase I/II detoxifying enzymes - DNA methylation	Lack of dose response relationship between exposure and biomarkers of effects
*Physical*		
Exercise	- anti-oxidative stress responses - mitochondrial activation - microRNA (myomiR)	Overweight prevention Cancer prevention Cardiovascular diseases prevention Metabolic syndrome and type II diabetes prevention
Heat	- Heat shock proteins	
Cold	- Mitochondria activation - Adipose brown tissue hyperplasia	Overweight prevention
Irradiation	- DNA repair - melanin neosynthesis	Skin cancer prevention
*Nutritional*		
Flavonoids	- anti-oxidative stress responses - anti-inflammatory stress responses - Nrf2	Cancer prevention
Polyphenols	- anti-oxidative stress responses - anti-inflammatory stress responses - phase I/II detoxifying enzymes - Nrf2	Cancer prevention
Caloric restriction/intermittent fasting	- autophagy - sirtuins	Increased life time span Decrease of spontaneous cancer incidence

**Table 2 ijms-21-07053-t002:** Overview of the key reviews or hypotheses concerning the role of epigenetic changes in adaptation to the environment.

Year	Main Topic	Main Conclusion	Ref.
2011	Revision of the link between hormesis and epigenetics.	Adaptive epigenetic rearrangements linking environmental factors can occur not only during early developmental stages but also through the adulthood, and they can cause hormesis.	[78]
2014	Discussion of the concept of epigenetic memory induced by developmental or environmental stimuli.	Three distinct paradigms of epigenetic memory (cellular, transcriptional, transgenerational) that operate on different time scales were suggested.	[67]
	Epigenetics in an ecotoxicological context.	The possibility of transgenerationally inherited, chemical stress-induced epigenetic changes with associated phenotypes. Epigenetically induced adaptation to stress upon long-term chemical exposure.	[79]
	Epigenetic memory and its potential to reflect previous stress exposure.	It is proposed that epigenetic “foot-printing” could identify classes of chemical contaminants to which organisms have been exposed throughout their lifetime. It is recommended that epigenetic mechanisms, alongside genetic mechanisms, should eventually be considered in environmental toxicity safety assessments and in biomonitoring studies.	[80]
2015	Focus on DNA methylation, emphasizing the aspects that could be relevant in human adaptations.	All genetic, epigenetic, and phenotypic variations are involved in human adaptation.	[81]
2017	Unusual results of the Czech biomonitoring studies (weak effect of exposure related to higher levels of environmental stressor) were revised.	Epigenetic adaptation via changes in DNA methylation pattern including impact of exposure history and their length were suggested as an explanation of unusual results. In addition, the epigenetic adaptation was suggested as a versatile mechanism related to various environmental stressors.	[42]
	Hypothesis related to epigenetic memory in response to environmental stressors.	Authors propose that an epigenetic memory can be established and maintained in self-renewing stem cell compartments.	[71]
2018	Context between low doses of environmental agents, adaptive response, epigenetic mechanisms, and toxicology research.	A beneficial effect resulting from activation of adaptive responses in the framework of hormesis was suggested. It should have a significant impact in biomedical/toxicological research.	[82]
2019	Evaluation of strategies of adaptation related to their speed.	Epigenetic switching was suggested as a quick strategy of adaptation to fluctuating environment.	[35]

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
