# Peer review of "The Molecular Mechanisms of Adaptive Response Related to Environmental Stress"

_ijms, 2020, doi:10.3390/ijms21197053_

Round 1
Reviewer 1 Report
I applaud the authors for their attempt at taking on such a muti-faceted, complex, and interesting line of research. However, they fail to deliver strong links with evidence-based reasoning and the organization of thoughts is redundant at times and confusing at others. This is a fascinating topic, but this review needs substantially more development as to reject it in its current form. Much of the MS seems to be purely speculative with very loose connections to the literature. Sections 1 to 3 ramble and do not link the evidence to assertions in a coherent manner. Section 4 is better and is interesting, but it lacks literature support. For a review, it is not nearly comprehensive enough - from a casual search, I found many unaccounted for articles on topics that are discussed. Such broad claims need to be substantiated with literature and there was a paucity of citations and missed literature, e.g.: https://www.sciencedirect.com/science/article/pii/S1568163711000055#:~:text=It%20appears%20likely%20that%20adaptive,ability%20of%20cells%20and%20organisms.) https://search.proquest.com/docview/1957857287?accountid=10906&rfr_id=info%3Axri%2Fsid%3Aprimo http://dx.doi.org/10.1289/ehp.1408459 https://link.springer.com/article/10.1007/s00204-014-1233-7 https://www.sciencedirect.com/science/article/pii/S1568163711000055#:~:text=It%20appears%20likely%20that%20adaptive,ability%20of%20cells%20and%20organisms https://www.ncbi.nlm.nih.gov/pmc/articles/PMC6213774/ https://stemcellsjournals.onlinelibrary.wiley.com/doi/full/10.1002/stem.2836 https://www.nature.com/articles/pr2007126 https://www.cell.com/trends/endocrinology-metabolism/fulltext/S1043-2760(19)30174-2
Author Response
Dear Editor:
thank you for your letter stating that our manuscript entitled: “The Adaptive Response to Environmental Stress: Significance in Preventive Medicine, Mechanisms and Role of the microRNA Machinery”
could be acceptable for publication in International Journal of Molecular Sciences pending revisions.
Accordingly, we have prepared a revised version of the manuscript acknowledging Referees’ and Editor’s comments as below specified:
Reviewer 1:
COMMENT 1. Sections 1 to 3 ramble and do not link the evidence to assertions in a coherent manner.
ANSWER 1. Sections 1 was completely re-written. Section 3 was deeply changed and rephrased. In both Sections redundant paragraphs were deleted. Assertions are now supported by newly added suitable references.
COMMENT 2. Section 4 is better and is interesting, but it lacks literature support. For a review, it is not nearly comprehensive enough - from a casual search, I found many unaccounted for articles on topics that are discussed.
ANSWER 2. 28 new references have been added as compared to the previous version to widen literature coverage as suggested. Furthermore, literature support related to section 4 (Epigenetic aspects of human adaptation to environmental pollution) was added in the newly added Table 2 now including 9 new references.
We sincerely thanks both reviewers and Editor for their efforts and suggestions that have really helped us to improve our manuscript that we hope could be now accepted for publication.
Sincerely,
Alessandra Pulliero
Reviewer 2 Report
This manuscript comprehensively summarized molecular mechanisms regarding adaptive responses. The manuscript is well-written and I enjoyed reading it. One minor issue is that the title is misleading. About the role of microRNA is only a small piece of this paper (just shortly introduced in Sec. 5). The authors should revise the title to fit with the contents.
Author Response
Dear Editor:
thank you for your letter stating that our manuscript entitled: “The Adaptive Response to Environmental Stress: Significance in Preventive Medicine, Mechanisms and Role of the microRNA Machinery”
could be acceptable for publication in International Journal of Molecular Sciences pending revisions.
Accordingly, we have prepared a revised version of the manuscript acknowledging Referees’ and Editor’s comments as below specified:
Reviewer 2:
COMMENT 1. The manuscript is well-written and I enjoyed reading it. One minor issue is that the title is misleading. About the role of microRNA is only a small piece of this paper (just shortly introduced in Sec. 5). The authors should revise the title to fit with the contents.
ANSWER 1. The title has been changed as requested.
We sincerely thanks both reviewers and Editor for their efforts and suggestions that have really helped us to improve our manuscript that we hope could be now accepted for publication.
Sincerely,
Alessandra Pulliero
Round 2
Reviewer 1 Report
Extensive additions of text and references make this a better manuscript. Needs a bit of cleanup for grammar, but is otherwise OK.